# DPG: EXPLOITING DATA AND PROCESS KNOWLEDGE FOR DIFFUSION GUIDANCE

## ABSTRACT

Diffusion models have excelled in imperfect-label guidance tasks, including weak-label guidance (e.g., style transfer) and degraded-label guidance (e.g., image super-resolution and deblurring), where supervision is incomplete or compromised. Current methods are either tailored to specific tasks, limiting generalization and transferability, or rely solely on loss-guided methods for imperfect-label tasks, imposing insufficient constraints and overlooking valuable domain priors, leading to suboptimal results. We argue that these limitations stem from the failure to uncover the universally applicable latent knowledge in these tasks and methods. To address this, we propose DPG, a universal framework that exploits data knowledge inherent to the tasks and process knowledge derived from reverse diffusion in imperfect-label guidance tasks. Explicitly, we diffuse imperfect-label data or its variants and then incorporate them into the initial stages of reverse diffusion, elevating the precision and efficiency of outputs in aligning with labels. As well, we capitalize on the temporal progression of the diffusion model's denoising path, ensuring that each step surpasses its predecessor in satisfying label constraints to refine optimization choice, thus improving guidance fidelity and accelerating convergence. By integrating this knowledge, DPG can achieve generalization and optimal performance in imperfect-label tasks, paving the way for future innovations in unified frameworks. Extensive experiments demonstrate DPG's effectiveness, consistently generating high-quality outputs.

## 1 INTRODUCTION

In recent times, diffusion models (DMs) Ho et al. (2020); Rombach et al. (2022) have become a leading generative framework due to their remarkable ability to model complex data distributions Nichol et al. (2021); Podell et al. (2023). Building on this, advanced guidance strategies, such as classifier-based guidance Dhariwal & Nichol (2021); Wallace et al. (2023) and classifier-free guidance Ho & Salimans (2022); Sanchez et al. (2023), have further refined the precision and flexibility of DMs, enabling more targeted control over the sampling process. With these improvements, DMs have been gradually applied to imperfect-label guidance tasks for outstanding performance, including weak-label and degraded-label guidance tasks, where supervision is incomplete or compromised.

Weak-label guidance tasks refer to scenarios characterized by incomplete or sparsely available label data information, thereby increasing the difficulty of learning reliable mappings between inputs and outputs. These scenarios frequently occur when exhaustive annotation is infeasible, as exemplified by text-guided generation Saharia et al. (2022); Zhang et al. (2023a), where textual prompts provide only high-level semantics, and style transfer Lei et al. (2025); Liu et al. (2023), where the desired style is implicitly specified rather than rigorously annotated. In this paper, we primarily focus on style transfer as a representative task. Existing style transfer methods generally fall into two categories: (1) Approaches based on learning feature mapping: These methods focus on learning mappings from diverse attributes to their respective feature spaces, as well as modeling the inter-relations among these feature spaces, e.g., T2I-Adapter Mou et al. (2024), StyleCrafter Liu et al. (2023), and StyleShot Gao et al. (2024). (2) Approaches based on prior feature exploitation: These methods exploit pre-trained models to extract necessary features and perform effective feature fusion using task-adaptive and context-aware fusion mechanisms without additional training, e.g.., StyleID Chung et al. (2024), Z* Deng et al. (2024), and InstantStyle Wang et al. (2024).

Degraded-label guidance tasks relate to scenarios where the available labels are degraded or low-fidelity derivatives of authentic data, produced through systematic distortion. These tasks inherently complicate the learning process by supplying incomplete and corrupted signals, with representative examples including image super-resolution Kim et al. (2025); Wu et al. (2024), where low-resolution images serve as degraded labels, image deblurring Zhang et al. (2022); Patel et al. (2024); Xu et al. (2025) based on blurred observations, and denoising Kawar et al. (2022); Li et al. (2024a) with noisy references. In this paper, we primarily focus on the first two tasks as representative tasks. Current degraded-label guidance methods can be broadly categorized into two classes: (1) Approaches based on strict constraints: These methods impose explicit constraints that enforce consistency between the generated data and the degraded labels, guiding the model to align its outputs with the provided supervision despite label imperfections, e.g., SITCOM Alkhouri et al. (2025), PSLD Rout et al. (2023), and DMAP Xu et al. (2025). (2) Approaches based on flexible sampling: These methods introduce adaptive mechanisms to relax the constraints of the traditional reverse diffusion pathway, guiding the generation process toward more flexible outputs while maintaining fidelity to the labels, e.g., FlowDPS Kim et al. (2025), FlowChef Patel et al. (2024), and DOC Li & Pereira (2024).

A unified framework has clear advantages over task-specific methods. Specifically, by combining different tasks, a unified framework encourages cross-fertilization of ideas. Strategies and insights from one task can positively influence others, leading to new and creative solutions. Besides, since these tasks use the same diffusion model, any improvements to the core components, e.g., sampling strategies, will enhance performance across all of them without needing to build and optimize separate architectures for each task. Thus, exploring the shared characteristics of different tasks to create a unified framework is valuable. Despite these advantages, several factors hinder the development of universally applicable methods: (1) The difference in data content is a major hurdle. In weak-label guidance tasks, the data provides only partial valid information, with the remainder being mostly irrelevant or detrimental. This makes it difficult to apply strong constraints, which is why these tasks often rely on feature extraction. In contrast, degraded-label guidance tasks have almost entirely valid information, making strong constraints more effective. (2) The misalignment of task objectives is a fundamental problem. Weak-label tasks prioritize visual quality and diversity, allowing for a range of plausible outputs. Degraded-label tasks, on the other hand, require a single, precise reconstruction that is faithful to the original target. Therefore, the difficulty in capturing and utilizing task-level commonalities poses a significant obstacle to the design of a unified framework.

Beyond the above methods, loss-guided approaches Ye et al. (2024); Yu et al. (2023); Castillo et al. (2025) can handle imperfect-label tasks. They tackle tasks by leveraging various loss-gradient guidance strategies or parameter configuration strategies. For example, TFG Ye et al. (2024) unifies loss-guidance methods and provides a parameter search strategy. FreeDom Yu et al. (2023) proposes a multi-conditional energy-function guidance method. These methods are a step toward unification, but they have some limitations. First, a loss function is often too coarse to fully guide complex tasks. Because it only provides a single numerical value to summarize a vast amount of information, it is blind to the valuable priors and granular details that exist within the data itself, making the guidance less accurate and leading to a suboptimal solution. For example, a loss function might indicate how far a stylized image is from the target, but it lacks the granular information needed to correct specific elements like textures, which exist only within the data itself and are not reducible to a mathematical expression. Besides, a loss-based optimization is executed sequentially, step-by-step. This process, which only guarantees local improvement at the current time step, is fundamentally susceptible to error propagation. Any bias or sub-optimality introduced in an earlier step will inevitably compound and accumulate over the entire sequence. This progressive cumulative error makes the overall optimization path inefficient, ultimately leading to slower convergence and a suboptimal final.

Building on the above observations, we introduce DPG, a unified approach that leverages task-inherent data knowledge and reverse diffusion-based process knowledge to address imperfect-label guidance tasks. Our motivation is twofold: First, while detailed, task-specific constraints can extract more information from labels, they reduce a method's generalizability. To create a framework that works across diverse data, we avoid explicit constraints. Instead, we diffuse the label data and integrate it early in the reverse diffusion process. This allows us to fully use the label content to make outputs more accurate and efficient without sacrificing flexibility. Second, adjusting a model's architecture or task objectives can improve performance, but it limits the method's ability to work with different models (like U-Net Rombach et al. (2022) or Transformer-based Peebles & Xie (2023)) and tasks. To overcome this, we apply a general constraint along the reverse diffusion path,

which is a common generative process for all diffusion models. We utilize the temporal structure of the reverse denoising process as a process knowledge of diffusion models to implement a progressive alignment mechanism, ensuring that the output at each timestep becomes increasingly aligned with the label compared to the previous one.

To our knowledge, this paper is the first study to analyze the gap between weak-label and degraded-label guidance tasks and to propose a unified approach to bridge it, paving the way for task-agnostic innovations. Overall, our main contributions are as follows: First, we analyze the intrinsic differences between weak-label and degraded-label guidance tasks and identify the key obstacles to generalization across them. Second, we introduce DPG, a unified framework that integrates task-inherent data knowledge and reverse diffusion-based process knowledge for imperfect-label guidance tasks, bridging the gap between the two tasks. Last, we conduct extensive experiments that validate the effectiveness and generalizability of DPG, achieving high-quality results on several specific tasks.

## 2 RELATED WORK

**Weak-label Guidance.** We focus on style transfer, generating images that preserve the content of a text or image while adopting the desired style of another image. Methods for this task can be divided into two categories: those that learn feature mappings and those that exploit prior features. The former primarily focuses on learn to map different attributes into their own feature spaces and model the relationships between them. For instance, StyleShot Gao et al. (2024) maps the style features extracted by a style-aware encoder into the CLIP Radford et al. (2021) space of a diffusion model. StyleCrafter Liu et al. (2023) trains a style feature extrator to map CLIP image features into a style feature space, refining and decoupling them from image features. CSGO Xing et al. (2024) establishes dual mappings from the CLIP image space to individual content and style representations. DEADiff Qi et al. (2024) maps image features into content and style spaces via a feature extractor, guided by textual prompts. StyleDrop Sohn et al. (2023) maps style attributes into the feature space via lightweight tuning, refined with human feedback. In contrast, prior feature exploitation methods utilize pre-trained models to extract key features. For example, InstanStyle Wang et al. (2024) derives style by subtracting the textual content embedding from the image embedding. StyleStudio Lei et al. (2025) obtains pure style features by decoupling content information extracted by ControlNet Zhang et al. (2023b) from style image features. StyleAlign Hertz et al. (2024) injects style by extracting a style image's queries, keys, and values, then aligns them in the attention layers.

Compared to previous methods, DPG directly uses the style image, adaptively extracting relevant style information. This approach can avoid biases that can arise from learning feature mappings and the incompleteness of feature extraction often found in pre-trained models.

**Degraded-label Guidance.** We focus on image super-resolution and image deblurring, reconstructing the original image from a low-resolution or blurred observation. Approaches to this task can be divided into two categories: methods based on strict constraints and methods based on flexible sampling. The former mainly uses explicit constraints to ensure consistency between the outputs and degraded labels, guiding the model to align with the given supervision despite label imperfections. For example, SITCOM Alkhouri et al. (2025) enforces consistency across multiple stages to make sure outputs align with labels and stay within a valid distribution. PSLD Rout et al. (2023) uses a correction constraint to keep solutions close to predictions. DMAP Xu et al. (2025) applies a consistency constraint, projecting the state onto a sphere. DCDP Li et al. (2024b) enforces consistency with lightweight methods like gradient descent, avoiding full backpropagation. Whereas, flexible sampling methods often reinterpret sampling as an alternative problem or deviate from the strict steps of the standard denoising process. FPS-SMC Dou & Song (2024) treats posterior sampling as a Bayesian filtering problem. DOC Li & Pereira (2024) turns the sampling process into a discretized optimal control problem. FlowChef Patel et al. (2024) uses vector field properties for goal-directed generation. FlowDPS Kim et al. (2025) prioritizes likelihood maximization early, then progressively relaxes constraints. In addition, task-specific methods, such as SeeSR Wu et al. (2024) and InvSR Yue et al. (2025) for super-resolution, have been thoroughly explored.

Compared with the above, DPG does not impose strict consistency constraints nor modify the sampling process. DPG utilizes data knowledge inherent to the tasks and process knowledge derived from reverse diffusion to tackle the degraded-label tasks, enabling more general exploration.

**Loss-gradient Guidance.** Some loss-gradient guidance methods can be applied to these tasks, focusing on the functioning of the loss gradient and the adjustment of parameters. TFG Ye et al. (2024) introduces a unified framework, consolidating existing methods and incorporating a gradient update rate strategy. FreeDoM Yu et al. (2023) redefines the loss as the energy, using gradient guidance to adjust the noise image towards the target, instead of directly modifying the output. AG Castillo et al. (2025) presents an adaptive mechanism that enhances efficiency by switching the conditioning step based on the cosine similarity between conditional and unconditional scores.

Unlike these methods, DPG uses intrinsic data knowledge as a prior, incorporating complete data information for high precision. Besides, DPG uses process knowledge from the reverse diffusion, emphasizing how data progressively aligns with the label, with each time step producing data that better matches the label to refine optimization choice, reducing error accumulation to some extent and improving output quality. In Fig. 3, process knowledge improves image quality in each step by re-choosing optimization points, causing sharp jumps or increased dynamics in the metric curve.

## 3 METHOD

### 3.1 REVISIT LATENT DIFFUSION MODELS

The latent diffusion models Rombach et al. (2022) work on the latent space. It has a forward diffusion that gradually adds noise to map data to a standard Gaussian distribution, and a reverse diffusion that steadily denoises to yield target-distribution data.

Thus, the clean latent vector $z_0$ can be first obtained:

$$z_0 = E(x), \tag{1}$$

where $E$ is an encoder of Rombach et al. (2022). $x$ denotes the clean data.

The forward diffusion is typically modeled as a Markov chain, where the latent vector $z_t$ at each state $t \in (1, T)$ is obtained by adding Gaussian noise to the clean latent vector $z_0$:

$$z_t = \sqrt{\bar{\alpha}_t} z_0 + \sqrt{1 - \bar{\alpha}_t} \epsilon, \tag{2}$$

where $\bar{\alpha}_t = \prod_{s=1}^{t}(1 - \beta_t), \beta_t \in (0, 1)$ adopts a fixed variance schedule. $\epsilon \sim \mathcal{N}(0, \mathbf{I})$ is the noise.

The reverse diffusion generates data by progressively denoising from a Gaussian noise sample $z_T \sim \mathcal{N}(0, \mathbf{I})$. Based on PLMS Liu et al. (2022), each preceding state $z_{t-1}$ is obtained:

$$z_{0|t} = \frac{z_t - \sqrt{1 - \bar{\alpha}_t}\, \epsilon_\theta(t)}{\sqrt{\bar{\alpha}_t}},$$

$$z_{t-1} = \sqrt{\alpha_{t-1}}\, z_{0|t} + \sqrt{1 - \alpha_{t-1} - \sigma_t^2}\epsilon_\theta(t) + \sigma_t z, \; z \sim \mathcal{N}(0, \mathbf{I}) \tag{3}$$

where $\epsilon_\theta(t) = \sum_{i=0}^{k-1} w_i \epsilon_\theta(z_{t-i}, t - i)$. $k = 4$ is for the historical steps. $w_i$ is the weighting coefficient. $\epsilon_\theta$ is predicted noise. U-Net Rombach et al. (2022) and DiT Peebles & Xie (2023) both can be used to predict noise, as the core principle of diffusion remains the same. For this paper, we use the U-Net as our foundational model. $\sigma_t^2 = 0$ for deterministic sampling.

Finally, the target image $x$ can be obtained via decoding:

$$x = D(z_0), \tag{4}$$

where $D$ represents the decoder of Rombach et al. (2022).

### 3.2 OVERALL FRAMEWORK OF DPG

We introduce DPG, which incorporates data and process knowledge, for imperfect-label guidance tasks (Fig. 1), with the algorithm in Sec. A of the Appendix.

**Data Knowledge Integration**. In imperfect-label tasks, all target information is derived from imperfect labels. To avoid information loss during learning network mapping or extracting features

Figure 1: The overall frameworks of DPG. Data knowledge is injected at the early time steps of the reverse diffusion process, while process knowledge is applied in the entire reverse diffusion process. Task-specific conditions, if any, are also maintained across the entire process.

by pre-trained models, we use the full information from the imperfect label as input, automatically extracting the useful information during the generation process, as shown in Fig. 1 and Fig. 2 (a).

In particular, given an imperfect label $y$, we first decide what operation $M$ is applied to $y$ to obtain a higher-quality initial label with more useful and informative content:

$$\hat{y} = M(y), \tag{5}$$

where $M$ is chosen based on the specific task, processing the label as needed to ensure the input information is well aligned with the task requirements. The detailed $M$ is in Sec. B of the Appendix.

Then, the processed $\hat{y}$ is used to obtain the initial latent vector $z_T$ at time step $T$ and denoising intermediate latent vector $\hat{c}_t$ at time step $t$ of the reverse diffusion process:

$$\hat{z} = E(\hat{y}),$$
$$z_T = \sqrt{\bar{\alpha}_T}\hat{z} + \sqrt{1 - \bar{\alpha}_T}\epsilon, \tag{6}$$
$$\hat{c}_t = \sqrt{\bar{\alpha}_t}\hat{z} + \sqrt{1 - \bar{\alpha}_t}\epsilon_{ti}, i = 1, \ldots, N_{iter1}$$

where the noise $\epsilon \sim \mathcal{N}(0, \mathbf{I})$. For $i = 1$, $\epsilon_{ti} = \epsilon$; otherwise, $\epsilon_{ti} = \epsilon_\theta(t)$ (Eq. 7). $N_{iter}$ is the number of iterations, as detailed in Sec. A of the Appendix. $\hat{c}_t$ is the noisy data containing the complete label information. Notably, without using label data directly, we inject noise, as only part of the information is useful, while the rest may be irrelevant or harmful. In style transfer, the style image content is unnecessary, while in super-resolution and deblurring, semantic information is needed, but details often contain noise. These irrelevant parts can reduce the generation quality. By adding noise and applying guidance, we let the model select the most relevant information for the task.

Subsequently, we inject imperfect-label data or its variants with noise into the early stages of reverse diffusion, improving the precision and efficiency of the outputs:

$$c_t = \alpha_{data} \times z_t + (1 - \alpha_{data}) \times \hat{c}_t,$$
$$\hat{\epsilon}_\theta(z_t, c_t, c_{task}) = \gamma_{data} \times \epsilon_\theta(c_t, c_{task})$$
$$+ (1 - \gamma_{data}) \times \epsilon_\theta(z_t, c_{task}), \tag{7}$$
$$\epsilon_\theta(t) = \hat{\epsilon}_\theta(z_t, \hat{c}_t, c_{task}),$$

where $z_t$ and $\hat{c}_t$ are combined by addition, allowing the information in $z_t$ to interact with that in $\hat{c}_t$, enabling adaptive adjustments to produce the most suitable outcome. $\alpha_{data}$ and $\gamma_{data}$ are weighting factors, with specific values provided in Sec. B of the Appendix. $c_{task}$ denotes the task-specific condition input: a text prompt for style transfer, and empty for image super-resolution and deblurring.

**Discussion.** SDEDIT Meng et al. (2021) generates an image from a user's guide, like a sketch, by adding noise at a given time step $t_0$ and then denoising from this starting point. Our DPG is fundamentally different from SDEdit. First, SDEdit only uses its abstract input as a "bridge" to indirectly access the model's knowledge. In contrast, DPG explicitly leverages the intrinsic knowledge of the input data. For example, we utilize fine-grained details like textures for style transfer and all available information for super-resolution and deblurring. Second, SDEdit's denoising begins from a fixed starting point, which is obtained by adding noise to the user's guide. DPG, however, guides every step of the denoising process by using rich data knowledge. This makes the path more

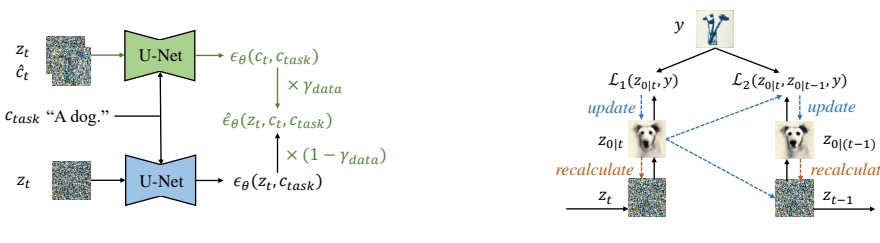

(a) Data Knowledge Integration  (b) Process Knowledge Integration

Figure 2: The detailed diagrams show the data and process knowledge integration in DPG.

direct and efficient, avoiding the "wobbly" or suboptimal solutions of SDEdit. Finally, SDEdit's use of data is not selective. Our DPG method, however, is adaptive, able to selectively use the most effective knowledge based on the predicted noise at the current time step, i.e., $\epsilon_{ti} = \epsilon_\theta(t)$ in Eq. 6.

We then obtain the predicted clean latent $z_{0|t}$:

$$z_{0|t} = \frac{z_t - \sqrt{1 - \bar{\alpha}_t}\, \epsilon_\theta(t)}{\sqrt{\bar{\alpha}_t}}, \tag{8}$$

where $z_{0|t}$ should align with $y$. Thus, we optimize $z_{0|t}$ to better capture the target information:

$$\begin{aligned}\mathcal{L}_1(z_{0|t}, y) &= f_{loss}(D(z_{0|t}), y), \\ z_{0|t} &= z_{0|t} - \eta_1 \times \nabla_{z_{0|t}} \mathcal{L}_1(z_{0|t}, y),\end{aligned} \tag{9}$$

where $f_{loss}$ is the task loss function. $\eta_1$ controls the step size of gradient updates to $z_{0|t}$, deciding the adjustment extent. More details are provided in Sec. B of the Appendix.

After that, we obtain the $z_{t-1}$ with rich information:

$$z_{t-1} = \sqrt{\alpha_{t-1}}\, z_{0|t} + \sqrt{1 - \alpha_{t-1} - \sigma_t^2}\, \epsilon_\theta(t) + \sigma_t z, \tag{10}$$

where $\sigma_t = 0$. Although Eq. 9 aligns $z_{0|t}$ with $y$, it doesn't ensure that the later prediction will be more accurate than the earlier due to cumulative error.

**Process Knowledge Integration.** In imperfect-label guidance tasks, relying on a pixel-level loss $\mathcal{L}_1$ has fundamental limitations. Because this loss provides only a static, final signal, it ignores the temporal evolution of the denoising process. This forces the model to re-optimize at each step, preventing a consistent and progressive optimization path. Simply put, this implies that the prediction $x_{0|t-1}$ is not obtained through a continuous optimization from $x_{0|t}$. Instead, their respective optimizations along the path are disconnected. Besides, the requirement for the model to optimize sequentially, step-by-step, renders this approach inherently susceptible to the problem of cumulative error. Since the optimization is locally focused, any minor bias or imprecision introduced in a preceding time step will inevitably propagate and compound across the entire subsequent sequence, leading to a suboptimal final result. Thus, we integrate process knowledge of reverse diffusion (Fig. 1 and Fig. 2 (b)), using the principle that each prediction should progressively align closer to the label than the previous step, eliminating cumulative error via incremental refinement and the selection of the optimal path:

$$\begin{aligned}\mathcal{L}_2 &= \max\left(\mathcal{L}_1(z_{0|t-1}, y) - \mathcal{L}_1(z_{0|t}, y) + \alpha_{margin}, 0\right), \\ z_{0|t-1} &= z_{0|t-1} - \eta_2 \times \nabla_{z_{0|t-1}} \mathcal{L}_2,\end{aligned} \tag{11}$$

where $\alpha_{margin}$ ensures that the negative sample $z_{0|t}$ stays at least a certain distance away from the positive sample $z_{0|t-1}$. $\eta_2$ governs the step size of gradient-based updates to $x_{0|t-1}$. Parameter values are in Sec. B of the Appendix. Leveraging the enhanced process prior, the predicted $z_{0|t-1}$ is expected to align more closely with the target than $z_{0|t}$. In the DMs, reverse diffusion reduces noise at each step. Fig. 3 illustrates the sharp inflection points and increased dynamics in the metric curve in our trajectory, reflecting the active path reselection central to generating higher-quality outputs.

Ultimately, we obtain the noisy data $z_{t-1}$, which incorporates precise and high-quality information:

$$z_{t-1} = \sqrt{\bar{\alpha}_{t-1}} z_{0|t-1} + \sqrt{1 - \bar{\alpha}_{t-1}} \epsilon. \tag{12}$$

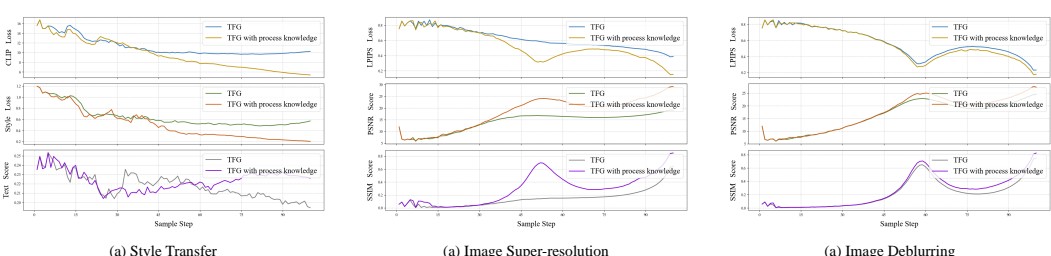

Figure 3: Process knowledge effect curve of DPG in imperfect-label guidance tasks.

By repeating the above steps in the reverse diffusion process, we obtain the final output:

$$x_0 = D(z_0), \tag{13}$$

where $x_0$ denotes data consistent with imperfect labels.

## 4 EXPERIMENTS

### 4.1 EXPERIMENT SETTINGS

We study imperfect-label guided tasks, adopting style transfer as the exemplar of weak-label guidance and super-resolution and deblurring as exemplars of degraded-label guidance. For style transfer, we compare various methods on 40,000 512×512 stylized images, created by combining 200 texts with 200 randomly sampled style images from the WikiArt Phillips & Mackintosh (2011). For super-resolution and deblurring, we evaluate on 1,000 randomly selected FFHQ Karras et al. (2019) images. For super-resolution, we downsample images by a factor of 4 and apply Gaussian noise with a standard deviation of 0.01, then generate 256×256 images. For deblurring, we use a Gaussian blur kernel with a size of 61 and a standard deviation of 3.0 and apply Gaussian noise with a standard deviation of 0.01 on 256×256 images. The detailed settings are provided in Sec. B of the Appendix.

### 4.2 COMPARISON

**Qualitative Comparisons.** We perform a qualitative comparison of the three tasks.

In Fig. 4 (a), current methods suffer from weak stylization (e.g., DEADiff, StyleDrop, TFG, and FreeDom), poor understanding of text prompts (e.g., StyleStudio, StyleCrafter, and CSGO), or content leakage of the style image (e.g., StyleShot, InstantStyle, and StyleAlign). In contrast, our DPG accurately captures text semantics (e.g., snow in the $2nd$ row and a white towel in the $3rd$ row) and achieves strong stylization (e.g., the dog in the $1st$ row and the fireworks in the $4th$ row).

In Fig. 4 (b), comparative methods show artifacts (e.g., PSLD and DMAP), poor photorealism (e.g., InvSR, FPS-SMC, and SITCOM), or blurriness/lack of detail (e.g., FlowDPS, FlowChef, and DOC), and outside the data distribution (e.g., TFG and FreeDom). DPG generates images without artifacts, presenting a clearer, more realistic, and visually layered appearance (e.g., the $1st$ and $2nd$ rows) and effectively restores fine details, e.g., the mole in the $3rd$ row and the temple hair in the $4th$ row.

In Fig. 4 (c), existing methods suffer from a biased distribution (e.g., DCDP, PSLD, FPS-SMC, SITCOM, FlowChef, and DOC), artifacts (e.g., DCDP, DMAP, and FlowDPS), and blurriness (e.g., FlowChef), outside the data distribution (e.g., TFG and FreeDom), whereas DPG produces images with a more accurate overall distribution (e.g., the background color in the $1st$ and $2nd$ rows) and effectively restores details (e.g., the folds of the hat in the $3rd$ row and the dimples in the $4th$ row).

Overall, DPG effectively uses data and process knowledge to achieve superior results.

**Quantitative Comparisons.** We also conduct a quantitative comparison of the three tasks.

For text-to-image style transfer, we evaluate with Text Score Sohn et al. (2023), Style Loss Huang & Belongie (2017), CLIP Loss Ye et al. (2024), and Preference Liu et al. (2021); Shang et al. (2025). Text Score is the cosine similarity between the stylized image and text CLIP embeddings, reflecting the semantic alignment with the text. Style Loss quantifies stylization via the mean and standard

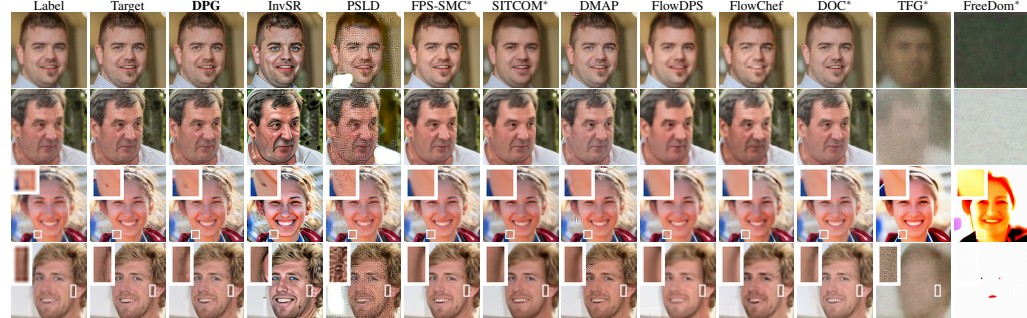

(a) Qualitative comparisons of text-to-image style transfer task.

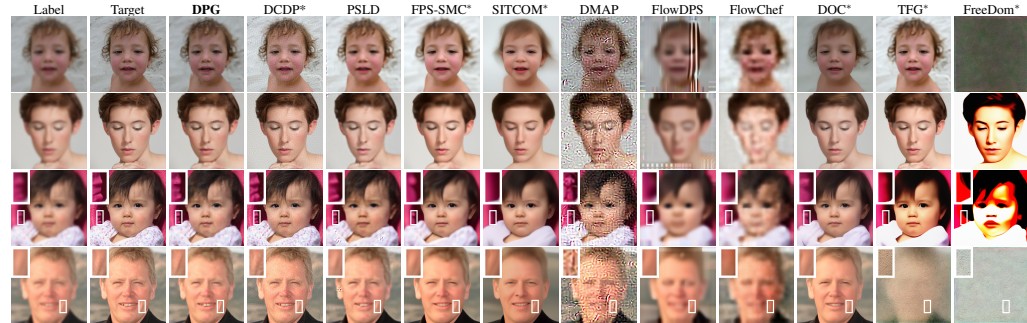

(b) Qualitative comparisons of image super-resolution task.

(c) Qualitative comparisons of image deblurring task.

Figure 4: Qualitative comparisons of the imperfect-label guidance task. An asterisk (∗) denotes models in the pixel space; those unmarked operate solely within the latent space.

| | DPG | StyleShot | StyleStudio | StyleCrafter | DEADiff | InstantStyle | StyleAlign | CSGO | StyleDrop | TFG | FreeDom |
|---|---|---|---|---|---|---|---|---|---|---|---|
| Text Score ↑ | 0.2952 | 0.2740 | 0.2852 | 0.2582 | 0.2907 | 0.1920 | 0.2553 | 0.2688 | 0.2780 | **0.3092** | 0.2933 |
| Style Loss ↓ | **0.6313** | 0.6747 | 0.9996 | 0.7444 | 1.7697 | 1.2691 | 0.8252 | 0.6793 | 1.3341 | 2.1120 | 2.7492 |
| CLIP Loss ↓ | **4.2334** | 6.0415 | 7.7899 | 7.0266 | 9.3485 | 8.4820 | 6.9670 | 6.5944 | 8.5434 | 7.7799 | 11.1131 |

(a) Quantitative comparisons of text-to-image style transfer task. The best results are in **bold**. The second best results are underlined

| | DPG | InvSR | PSLD | FPS-SMC | SITCOM | DMAP | FlowDPS | FlowChef | DOC | TFG | FreeDom |
|---|---|---|---|---|---|---|---|---|---|---|---|
| PSNR Score ↑ | **28.8600** | 21.5851 | 24.9210 | 27.6305 | 25.3733 | 26.3395 | 25.0459 | 23.1086 | 26.7617 | 26.3441 | 10.7963 |
| SSIM Score ↑ | 0.8233 | 0.6713 | 0.6397 | **0.8283** | 0.7805 | 0.7840 | 0.7438 | 0.7458 | 0.8167 | 0.7979 | 0.2546 |
| LPIPS Loss ↓ | **0.1573** | 0.2374 | 0.2560 | 0.2029 | 0.2460 | 0.2499 | 0.3114 | 0.2868 | 0.1717 | 0.2106 | 0.7190 |

(b) Quantitative comparisons of image super-resolution task. The best results are in **bold**. The second best results are underlined

| | DPG | DCDP | PSLD | FPS-SMC | SITCOM | DMAP | FlowDPS | FlowChef | DOC | TFG | FreeDom |
|---|---|---|---|---|---|---|---|---|---|---|---|
| PSNR Score ↑ | 27.5794 | **27.9110** | 25.8065 | 25.7486 | 23.0995 | 18.8024 | 21.1828 | 20.5104 | 25.2189 | 22.6209 | 12.3003 |
| SSIM Score ↑ | **0.7736** | 0.7384 | 0.7566 | 0.7665 | 0.7082 | 0.3096 | 0.5987 | 0.6067 | 0.7362 | 0.7149 | 0.3105 |
| LPIPS Loss ↓ | **0.2236** | 0.2325 | 0.2675 | 0.2540 | 0.3100 | 0.5541 | 0.4887 | 0.4934 | 0.2448 | 0.2869 | 0.6764 |

(c) Quantitative comparisons of image deblurring task. The best results are in **bold**. The second best results are underlined

Table 1: Quantitative comparisons of the imperfect-label guidance task.

deviation loss of VGG features between the stylized and the style images. CLIP Loss evaluates stylization via the Gram loss of CLIP embeddings of the stylized and style images. In Tab. 1 (a), our method attains the second highest Text Score, the lowest Style Loss, and the lowest CLIP Loss,

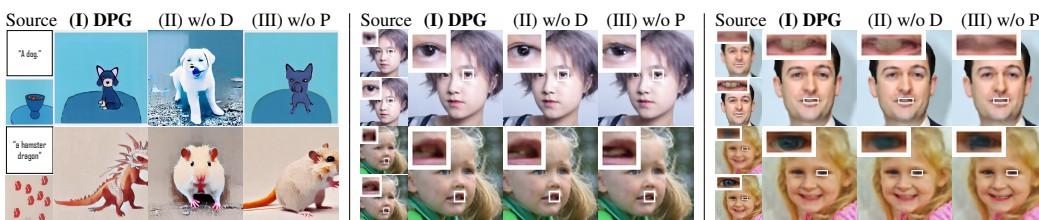

(a) Style transfer ablation study.  (b) Super-resolution ablation study.  (c) Deblurring ablation study.

Figure 5: Qualitative ablation results. "w/o D (P)" denotes "without data (Process) knowledge".

|  | (I) DPG | (II) w/o D | (III) w/o P |  | (I) DPG | (II) w/o D | (III) w/o P |  | (I) DPG | (II) w/o D | (III) w/o P |
|---|---|---|---|---|---|---|---|---|---|---|---|
| Text Score ↑ | 0.2952 | 0.2943 | **0.3008** | PSNR ↑ | **0.6313** | 28.8155 | 28.7759 | PSNR ↑ | **4.2334** | 27.5188 | 26.8616 |
| Style Loss ↓ | **0.6054** | 0.8098 | 0.9201 | SSIM ↑ | **0.8233** | 0.8224 | 0.8148 | SSIM ↑ | **0.7736** | 0.7711 | 0.7496 |
| CLIP Loss ↓ | **4.0579** | 4.7909 | 5.2108 | LPIPS ↓ | **0.1573** | 0.1574 | 0.1818 | LPIPS ↓ | **0.2236** | 0.2241 | 0.2590 |

(a) Style transfer ablation study.  (b) Super-resolution ablation study.  (c) Deblurring ablation study.

Table 2: Quantitative ablation results. "w/o D (P)" denotes "without data (Process) knowledge".

showing that our stylized images are the most faithful to the text prompts and the most strongly stylized. Despite TFG's marginal lead in Text Score, its Style and CLIP Losses are substantially higher. This underscores our method's superior style consistency and text-image fidelity.

For super-resolution and deblurring, we evaluate using PSNR Score Alkhouri et al. (2025), SSIM Score Patel et al. (2024), and LPIPS Loss Xu et al. (2025). PSNR and SSIM evaluate how faithfully the generated image preserves the target image through the peak signal-to-noise ratio and structural similarity, respectively. LPIPS applies perceptual networks to assess semantic and structural differences between the generated image and the target image. In Tab. 1 (b), our method has the highest PSNR Score and the lowest LPIPS Loss. While its SSIM is slightly lower than FPS-SMC, the latter shows much higher LPIPS Loss. In Tab. 1 (c), our method achieves the highest SSIM Score and the lowest LPIPS Loss, with PSNR slightly below DCDP. These demonstrate our method's superior fidelity and perceptual quality in the image super-resolution and deblurring tasks.

Overall, DPG exploits both data and process knowledge to achieve superior accuracy and robustness.

### 4.3 ABLATION STUDY

**Data Knowledge.** We exclude the data knowledge injection component to assess its impact. As shown in columns (I) and (II) of Fig. 5, the stylized images in (II) show more of the original model's biases, e.g., the white dog in the $1st$ row. The super-resolved images in (II) lack details like the double eyelids in the $1st$ row, while the deblurred images in (II) lack clarity, especially the eyes in the $2nd$ row. Moreover, the quantitative results confirm that data knowledge injection is effective in Tab. 2, highlighting its positive impact.

**Process Knowledge.** We remove the process knowledge to evaluate its effect. For columns (I) and (III) of Fig. 5, the images in (III) show style biases, e.g., the colors of the dog in the $1st$ row and the "hamster dragon" in the $2nd$ row. The super-resolved images in (III) have inaccurate details, like the teeth in the $2nd$ row. The deblurred images in (III) show limited detail recovery, e.g., the eyes in the $2nd$ row. The results in Tab. 2 also confirm that process knowledge is both essential and effective.

### 5 CONCLUSION

We unify the weak-label guidance task and degraded-label guidance task into the imperfect-label guidance task. Next, we provide a comprehensive analysis of the rationale behind this task unification, its benefits, and the challenges that hinder its practical implementation. To address these issues, we introduce DPG, a general framework that integrates task-specific data knowledge and diffusion-model process knowledge. DPG bridges the gap between the two task types and lays the groundwork for unified solutions. Extensive experiments on style transfer, image super-resolution, and image deblurring confirm the efficacy of our framework.

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

# APPENDIX

## A  DPG ALGORITHM

To offer a clearer understanding of our approach, we provide a detailed algorithm in Alg. 1. This illustrates the integration of data knowledge inherent to the specific task and process knowledge from reverse diffusion, and how they collaborate to solve the imperfect-label guidance task.

---

**Algorithm 1** DPG: Exploiting Data and Process Knowledge for Diffusion Guidance

---

1: **Input:** Diffusion model $\epsilon_\theta$, encoder $E$, decoder $D$, label $y$, label operation $M$, evaluation function $f_{loss}$, task condition $c_{task}$
2: $\hat{y} = M(y), \hat{z} = E(\hat{y})$
3: $z_T = \sqrt{\bar{\alpha}_T}\hat{z} + \sqrt{1 - \bar{\alpha}_T}\epsilon$, where $\epsilon \sim \mathcal{N}(0, \mathbf{I})$
4: **for** $t = T, \cdots, T_1$ **do**
5:     **for** $i = 1, \cdots, N_{iter1}$ **do**
6:         $\hat{c}_t = \sqrt{\bar{\alpha}_t}\hat{z} + \sqrt{1 - \bar{\alpha}_t}\epsilon$, where $\epsilon \sim \mathcal{N}(0, \mathbf{I})$ when $i = 1$
7:         $c_t = \alpha_{data} \times z_t + (1 - \alpha_{data}) \times \hat{c}_t$
8:         $\hat{\epsilon}_\theta(z_t, c_t, c_{task}) = \gamma_{data} \times \epsilon_\theta(c_t, c_{task}) + (1 - \gamma_{data}) \times \epsilon_\theta(z_t, c_{task})$
9:         $\epsilon_\theta(t) = \hat{\epsilon}_\theta(z_t, \hat{c}_t, c_{task})$
10:        $z_{0|t} = (z_t - \sqrt{1 - \bar{\alpha}_t}\,\epsilon_\theta(t))/\sqrt{\bar{\alpha}_t}$
11:        $z_{0|t} = z_{0|t} + \eta_1 \times \nabla_{z_{0|t}}\mathcal{L}_1(z_{0|t}, y)$, where $\mathcal{L}_1(z_{0|t}, y) = f_{loss}(D(z_{0|t}), y)$
12:        $z_t = z_{0|t}\sqrt{\bar{\alpha}_t} + \sqrt{1 - \bar{\alpha}_t}\epsilon_\theta(t), \epsilon = \epsilon_\theta(t)$
13:     **end for**
14:     $z_{t-1} = \sqrt{\bar{\alpha}_{t-1}}z_{0|t} + \sqrt{1 - \bar{\alpha}_{t-1}}\epsilon_\theta(t)$
15:     **for** $j = 1, \cdots, N_{iter2}$ **do**
16:         $\hat{c}_{t-1} = \sqrt{\bar{\alpha}_{t-1}}\hat{z} + \sqrt{1 - \bar{\alpha}_{t-1}}\epsilon$, where $\epsilon \sim \mathcal{N}(0, \mathbf{I})$ when $j = 1$
17:         $c_{t-1} = \alpha_{data} \times z_{t-1} + (1 - \alpha_{data}) \times \hat{c}_{t-1}$
18:         $\hat{\epsilon}_\theta(z_{t-1}, c_{t-1}, c_{task}) = \gamma_{data} \times \epsilon_\theta(c_{t-1}, c_{task}) + (1 - \gamma_{data}) \times \epsilon_\theta(z_{t-1}, c_{task})$
19:         $\epsilon_\theta(t - 1) = \hat{\epsilon}_\theta(z_{t-1}, \hat{c}_{t-1}, c_{task})$
20:         $z_{0|t-1} = (z_{t-1} - \sqrt{1 - \bar{\alpha}_{t-1}}\,\epsilon_\theta(t-1))/\sqrt{\bar{\alpha}_{t-1}}$
21:         $\mathcal{L}_2 = \max\left(\mathcal{L}_1(z_{0|t-1}, y) - \mathcal{L}_1(z_{0|t}, y) + \alpha_{margin}, 0\right)$
22:         $z_{0|t-1} = z_{0|t-1} + \eta_2 \times \nabla_{z_{0|t-1}}\mathcal{L}_2$
23:         $z_{t-1} = z_{0|t-1}\sqrt{\bar{\alpha}_{t-1}} + \sqrt{1 - \bar{\alpha}_{t-1}}\epsilon_\theta(t), \epsilon = \epsilon_\theta(t-1)$
24:     **end for**
25: **end for**
26: **for** $t = T_1 - 1, \cdots, 1$ **do**
27:     **for** $i = 1, \cdots, N_{iter1}$ **do**
28:         $\epsilon_\theta(t) = \hat{\epsilon}_\theta(z_t, c_{task})$,
29:         $z_{0|t} = (z_t - \sqrt{1 - \bar{\alpha}_t}\,\epsilon_\theta(t))/\sqrt{\bar{\alpha}_t}$
30:         $z_{0|t} = z_{0|t} + \eta_1 \times \nabla_{z_{0|t}}\mathcal{L}_1(z_{0|t}, y)$, where $\mathcal{L}_1(z_{0|t}, y) = f_{loss}(D(z_{0|t}), y)$
31:         $z_t = z_{0|t}\sqrt{\bar{\alpha}_t} + \sqrt{1 - \bar{\alpha}_t}\epsilon_\theta(t)$
32:     **end for**
33:     $z_{t-1} = \sqrt{\bar{\alpha}_{t-1}}z_{0|t} + \sqrt{1 - \bar{\alpha}_{t-1}}\epsilon_\theta(t)$
34:     **for** $j = 1, \cdots, N_{iter2}$ **do**
35:         $\epsilon_\theta(t - 1) = \hat{\epsilon}_\theta(z_{t-1}, c_{task})$,
36:         $z_{0|t-1} = (z_{t-1} - \sqrt{1 - \bar{\alpha}_{t-1}}\,\epsilon_\theta(t-1))/\sqrt{\bar{\alpha}_{t-1}}$
37:         $\mathcal{L}_2 = \max\left(\mathcal{L}_1(z_{0|t-1}, y) - \mathcal{L}_1(z_{0|t}, y) + \alpha_{margin}, 0\right)$
38:         $z_{0|t-1} = z_{0|t-1} + \eta_2 \times \nabla_{z_{0|t-1}}\mathcal{L}_2$
39:         $z_{t-1} = z_{0|t-1}\sqrt{\bar{\alpha}_{t-1}} + \sqrt{1 - \bar{\alpha}_{t-1}}\epsilon_\theta(t)$
40:     **end for**
41: **end for**
42: $x_0 = D(z_0)$
43: **Output:** Sample $x_0$

---

## B  DETAILED EXPERIMENT SETTINGS

To enhance the understanding of our experiments, we provide detailed task-specific settings, enabling others to replicate our results accurately.

### B.1 STYLE TRANSFER

**Label Operation** In style transfer tasks, the objective is to generate the style of the reference image, a highly abstract concept that is difficult to initialize well. Thus, no operation is needed for the label. That is $M = None$.

**Loss Function.** Similar to Gatys et al. (2016); Ye et al. (2024); Huang & Belongie (2017), we use statistical feature loss to preserve the style of the style image $y$.

*VGG Loss.* VGG Simonyan & Zisserman (2014) captures the basic shapes and textures of the image, ensuring pixel-level similarity in fine details. Therefore, we use VGG loss to align the detailed features of the generated stylized image with the style image:

$$
\mathcal{L}_{VGG} = \sum_{i=1}^{5} (||\mu(\phi_i(D(z_{0|t}))) - \mu(\phi_i(y))||_2 \\
+ ||\sigma(\phi_i(D(z_{0|t}))) - \sigma(\phi_i(y))||_2),
\tag{14}
$$

where $z_{0|t}$ is the predicted clean stylized image at time step $t$, given $c_{task}$ as the text content image and $y$ as the style image. $\phi_i$ is the feature map extracted from layer $Relui\_1$ of a pre-trained VGG-19 network Simonyan & Zisserman (2014). $\mu$ and $\sigma$ are channel-wise mean and standard deviation, respectively.

*CLIP Loss.* CLIP Radford et al. (2021) can understand the high-level semantics of images and provides comprehensive image information. Thus, we use the Gram matrix Gatys et al. (2016) from CLIP's image encoding to align the overall style of the style image:

$$
\mathcal{L}_{CLIP} = ||G(CLIP(D(z_{0|t}))) - G(CLIP(y))||_2,
\tag{15}
$$

where $G$ is the Gram matrix computation operation Gatys et al. (2016).

*Total Loss.* The final loss can be obtained:

$$
\mathcal{L}_1(z_{0|t}, y) = \mathcal{L}_{CLIP} + \alpha_{loss} \times \mathcal{L}_{VGG},
\tag{16}
$$

where $\alpha_{loss}$ represents the contribution of the CLIP loss to the total loss, with $\alpha_{loss} = 0.02$.

**Parameter Configuration.** DPG is implemented using the SDM1.4 Rombach et al. (2022) as the foundation. The sampling process is configured to 50 steps, ensuring both high quality and fast generation of stylized images. The mixing weight of the latent vector is set to $\alpha_{data} = 0.12$, which controls the degree of style information blending in the generated results. The mixing weight for label-conditioned noise is set to $\gamma_{data} = 0.2$, controlling the influence of style information from the reference style image on the denoising process, thus regulating the style transfer effect. To achieve a balance between the quality and efficiency of the generated results, we set the number of iterative updates for predicting clean data at the current and next time steps to $N_{iter1} = 3$ and $N_{iter2} = 3$, respectively. A gradually deteriorating parameter strategy Ye et al. (2024) is used to set the iterative update rates $\eta_1$ and $\eta_2$ for predicting clean data at the current and next time steps, with an initial update rate of 2.0. To effectively incorporate data knowledge, enhance style information, and prevent content leakage from the style image, we inject data knowledge during the first 18 time steps of the reverse diffusion, thereby generating highly stylized images that are consistent with the text. For incorporating process knowledge, we set $\alpha_{margin} = 1.0$ to calibrate the minimum distance between the label data $y$ and the positive sample $z_{0|t-1}$, as well as the negative sample $z_{0|t}$.

### B.2 IMAGE SUPER-RESOLUTION

**Label Operation.** Since label data $y$ is obtained via downsampling, we apply a simple interpolation function from Torch Paszke et al. (2017) for upsampling by the corresponding factor to ensure efficiency, thereby obtaining the initialized label data. Notably, a better upsampling method theoretically improves the performance of our DPG. That is $M = upsampling$.

**Loss Function.** Similar to Zhang et al. (2025); Chung et al. (2022), we apply the Euclidean loss to align the generated image with the label image $y$:

$$
\mathcal{L}_1(z_{0|t}, y) = ||M(D(z_{0|t})) - y||_2,
\tag{17}
$$

where $y$ is the noisy downsampled label. $M$ represents the downsampling degradation operation.

**Parameter Configuration.** The weights employed by DPG are pre-trained Rout et al. (2023) on the FFHQ dataset Karras et al. (2019). The sampling process is set to 200 steps, balancing the generation speed and the quality of high-resolution images. The latent vector's mixing weight is set to $\alpha_{data} = 0.12$, adjusting the degree to which degraded label information is incorporated into the high-resolution output. The mixing weight for label-conditioned noise is set to $\gamma_{data} = 0.1$, integrating as many content details as possible from the downsampled noisy label data in the denoising process, thereby regulating the quality of the restored image. To balance the quality and efficiency of the generated results, we set the number of iterative updates for predicting clean data at the current and subsequent time steps to $N_{iter1} = 3$ and $N_{iter2} = 3$, respectively. The AdamW Loshchilov & Hutter (2017) algorithm is employed to automatically determine the optimal optimization update rate. The update rate for predicting clean data at the current time step is divided into three phases: the first accelerates convergence, the second stabilizes it, and the third prevents oscillations. Specifically, for reverse diffusion steps 0 to 130, we set $\eta_1 = 0.08$; for steps 130 to 160, $\eta_1 = 0.015$; and for steps 160 to 200, $\eta_1 = 0.006$. The update rate for predicting clean data at the next time step is initialized to $\eta_2 = 0.02$, ensuring stable convergence. To inject data knowledge, given that the degraded label data contains only coarse semantic information and lacks high-frequency details, we inject data knowledge only during the first 8 time steps of the reverse diffusion process to expedite the generation of specific semantics and content structure in the image. For integrating process knowledge into the diffusion model, we set $\alpha_{margin} = 1.0$ to regulate the minimum distance between the label data $y$, the positive sample $z_{0|t-1}$, and the negative sample $z_{0|t}$.

### B.3 IMAGE DEBLURRING

**Label Operation.** As our degraded label data $y$ is generated via Gaussian blurring, we apply a WienerDeblur Wiener (1964) for deblurring to obtain the initialized label data. It is worth noting that employing a more effective deblurring method could theoretically improve the performance of our DPG. That is $M = WienerDeblur$.

**Loss Function.** Similar to Zhang et al. (2025); Chung et al. (2022), we apply the Euclidean loss to align the generated image with the label image $y$:

$$\mathcal{L}_1(z_{0|t}, y) = ||M(D(z_{0|t})) - y||_2, \tag{18}$$

where $y$ is the noisy blurring label data. $M$ represents the blurring degradation operation.

**Parameter Configuration.** The weights utilized by DPG are pre-trained Rout et al. (2023) on the FFHQ dataset Karras et al. (2019). The sampling steps are configured to 200 steps, effectively balancing the trade-off between generation speed and the quality of the deblurred images. The mixing weight for the latent vector, $\alpha_{data} = 0.12$, regulates the extent to which degraded label information influences the blurred output. The label-conditioned noise mixing weight, $\gamma_{data} = 0.1$, determines the extent to which content details from the noisy, blurred label data are incorporated into the denoising process, thereby controlling the quality of the output deblurred image. To trade off between quality and efficiency, we set the iterative updates for predicting clean data at both the current and the next time steps to $N_{iter1} = 3$ and $N_{iter2} = 4$, respectively. We adopt the AdamW Loshchilov & Hutter (2017) optimizer to adaptively adjust the optimal update rate during generation. The update rate for predicting clean data at the current time step is scheduled in three phases: an initial phase to accelerate convergence, a middle phase to stabilize it, and a final phase to suppress oscillations. Specifically, for reverse diffusion steps 0 to 130, we set $\eta_1 = 0.13$; for steps 130 to 150, $\eta_1 = 0.015$; and for steps 150 to 200, $\eta_1 = 0.006$. To ensure stable convergence, the update rate for predicting clean data at the next time step is specified as $\eta_2 = 0.02$. Because degraded label data carries mainly coarse semantic information and minimal high-frequency details, we restrict data knowledge injection to the first 50 steps of reverse diffusion, facilitating faster emergence of semantic content and structure. In integrating the knowledge of the process, $\alpha_{margin} = 1.0$ controls the minimum distance separating the label data $y$, the positive sample $z_{0|t-1}$, and the negative sample $z_{0|t}$.

### B.4 PARAMETER SELECTION STRATEGY

$T_1$: The parameter determines how many initial steps of the reverse diffusion process are used to inject data knowledge. The choice of $T_1$ depends on the specific task and the quality of the information provided by the processed labels. For example, in style transfer tasks, an excessively large $T_1$

can lead to content leakage from the style image, while a value that is too small will result in a less stylized output. In image super-resolution, a large $T_1$ might incorporate too many blurred details into the model, thereby degrading its super-resolution capabilities. Conversely, a small $T_1$ limits the available knowledge and also restricts the model's performance. In image deblurring, a large $T_1$ could inject biases from simple deblurring algorithms into the generation process, causing the deblurred images to inherit these specific biases. A small $T_1$, on the other hand, would underutilize the structural detail knowledge, thus reducing the model's deblurring ability. Therefore, for these three tasks, our selection of $T_1$ is a balance between theoretical considerations and practical experience.

$\boldsymbol{N_{iter1}}$: The parameter represents the number of iterative updates applied to the predicted data at the current timestep during the reverse diffusion process. The choice of $N_{iter1}$ depends on the specific task, whether the updated data remains within the model's expected data distribution, and the overall effectiveness of the updates. For instance, in style transfer, since we primarily learn the overall style and detailed textures rather than the exact content of the style image, we can appropriately reduce the number of iterations. Conversely, in super-resolution and deblurring tasks, the model needs to recover specific semantics, layouts, and content details from the labels, so we can increase the number of iterations to improve performance. It's important to note that excessively increasing the number of iterations is akin to an optimization process. This can cause the updated data distribution to significantly deviate from the initial predicted distribution, often resulting in the creation of artificial artifacts. Furthermore, in the early stages of reverse diffusion, the amount of noise that needs to be reintroduced is substantial. At this point, increasing the number of iterations offers little utility in providing label-compliant information and is extremely time-consuming. Therefore, our selection of $N_{iter1}$ is a balance between theoretical considerations and practical experience.

$\boldsymbol{\eta_1}$: The parameter controls the speed of iterative updates to the predicted data at the current timestep during the reverse diffusion process. The magnitude of the learning rate depends on the required optimization granularity and the stability of the optimization for a specific task. For instance, in style transfer, since we are concerned with the overall feature distribution rather than the semantic content of the style image, the optimization is relatively coarse. Thus, we can use a higher learning rate. Conversely, for super-resolution and deblurring, where optimization requires attention to specific layouts and structural details, a moderate learning rate is more appropriate. A learning rate that is too high can cause the optimization to oscillate, while one that is too low may not be sufficient. Furthermore, because optimization for super-resolution and deblurring initially focuses on the overall layout and then on specific content and structural details, our learning rate should be gradually decreased. This allows for a more stable refinement of fine details in later stages. If the $\eta_1$ remains too high and causes oscillations at this point, the generated details could be significantly different. Based on theoretical insights and practical experience, we use a relatively fixed $\eta_1$ for style transfer, while super-resolution and deblurring tasks employ a phased, gradually decreasing $\eta_1$.

$\boldsymbol{N_{iter2}}$: The parameter represents the number of iterative updates applied to the predicted data for the next timestep during the reverse diffusion process. The selection of $N_{iter2}$ depends on the effectiveness of the updates and whether the updated data distribution remains within the model's valid range. The primary goal of this update step is to widen the loss gap between the predicted data and the label data for the current timestep and the next one. In other words, we want the loss for the next timestep to be significantly smaller than the loss for the current one. For this reason, we can appropriately increase the number of iterations. However, a higher number of iterations raises concerns about the effectiveness of the updates and the validity of the data distribution. In the early stages of reverse diffusion, an excessive number of iterations reduces efficiency, and there is no guarantee that the updated data distribution will stay within the model's valid range. Therefore, based on the theoretical analysis and practical experience mentioned above, we have made a reasonable choice for the $N_{iter2}$ value across the three tasks, balancing both efficiency and effectiveness.

$\boldsymbol{\eta_2}$: The parameter controls the speed of iterative updates for the predicted data at the next timestep during the reverse diffusion process. The learning rate is determined by the stability of the optimization and whether the updated data distribution remains within the model's valid range. Our objective for this optimization step is to ensure that the loss between the next timestep's predicted data and the label is smaller than the loss from the current timestep. We also want the optimization to remain stable and the updated data to stay within a reasonable range. Consequently, we must fine-tune the next timestep's predicted data within a narrow, sensible range, as it cannot be set too high. An excessively high $\eta_2$ can cause the optimization to overshoot the optimal solution, leading to oscillations where the loss value bounces erratically instead of steadily decreasing. This is analogous to taking

giant, clumsy steps toward a destination, often overshooting the mark and having to backtrack. Such instability can prevent the model from converging to a valid solution, or it might cause the updated data to fall outside the model's expected distribution, generating nonsensical or artefactual results. Conversely, a learning rate that is too low can lead to under-optimization. The model's updates would be too small and gradual, making the convergence process exceedingly slow and potentially preventing the model from ever reaching an optimal state within a reasonable timeframe. Therefore, our selection of $\eta_2$ is a balance between theoretical considerations and practical experience.

$\boldsymbol{\alpha_{data}}$: The parameter governs the degree of mixing between data knowledge and the data at the current timestep. The selection of $\alpha_{data}$ depends on the specific task and the amount of valid information that can be provided by the processed labels. Specifically, for style transfer tasks, our focus is on the style of the image rather than its content. Therefore, we can appropriately reduce the value of $\alpha_{data}$ during mixing to lessen the blend of content information. For super-resolution and deblurring tasks, if the quality of the label data obtained through label operations is high, we can use a slightly higher value of a to learn more semantics and details. Conversely, if the quality of the label data is low but we still use a high mixing degree, the final generated image quality will be compromised due to the injection of too much low-quality knowledge. Ultimately, the choice of $\alpha_{data}$ is a critical balancing act to ensure that the model absorbs the right amount of information without being misled by noisy or irrelevant data, directly impacting the quality and fidelity of the final output. The parameter acts as a dial, allowing us to precisely control the influence of external knowledge on the generative process. Therefore, based on the theoretical analysis and practical experience mentioned above, we have made a reasonable choice for the parameter $\alpha_{data}$ across these three tasks.

$\boldsymbol{\gamma_{data}}$: The parameter regulates the proportion of data knowledge injected throughout the entire generation process. It is analogous to the classifier-free guidance scale in conditional diffusion models. However, instead of using noise predicted with and without a text condition, we use noise predicted with and without a knowledge injection to guide the generation. The selection of $\gamma_{data}$ depends on the specific task and the quality of the labels obtained after preprocessing. For style transfer tasks, we only require the style of the image, not its content. Therefore, the value of $\gamma_{data}$ should not be excessively large, as this could lead to content leakage from the style image. For super-resolution and deblurring tasks, the higher the quality of the preprocessed labels, the greater the proportion of data knowledge should be. This means a higher value for $\gamma_{data}$ is needed to achieve a higher-quality generated image. Therefore, based on the theoretical analysis and practical experience discussed above, we have made a reasonable choice for the parameter $\gamma_{data}$.

$\boldsymbol{\alpha_{margin}}$: The parameter controls the maximum possible reduction in the loss between the predicted data and the label from the current timestep to the next. The value of this parameter is closely tied to the magnitude of the loss itself. Specifically, for style transfer tasks, where we focus on overall style, the loss is typically on a larger order of magnitude. Thus, we can set a higher value for $\alpha_{margin}$ to allow for a more significant decrease in loss. Conversely, in super-resolution and deblurring tasks, the emphasis is on precise detail restoration, and the loss magnitude is generally smaller. Consequently, a smaller value for $\alpha_{margin}$ is more appropriate to ensure fine-grained and stable optimization. Based on these theoretical considerations and practical experience, we have made a reasonable choice for the $\alpha_{margin}$ parameter across all three tasks.

## C   FUTURE WORK

### C.1   ADAPTIVE PARAMETER SELECTION STRATEGY

Based on our analysis of the parameter selection strategies in Sec. B.4, we find that the choice of parameters is closely tied to the data distribution during optimization, the stability of the optimization process, and the specific task. Given that the entire optimization is driven by the loss between the predicted data and the label data, we can leverage this core mechanism. This insight leads us to propose a loss-adaptive parameter selection strategy. This strategy will dynamically adjust relevant parameters based on the changing loss between the predicted and label data. By adopting this approach, we will achieve optimal parameter configurations, leading to the generation of higher-quality and more accurate results. This method transforms parameter selection from a static, manual decision based on human experience into a dynamic, intelligent process guided by real-time loss feedback, significantly enhancing the model's performance and generalization ability. This will be a significant direction for our future research.

## C.2 Accelerated Inference

Our method involves certain optimization loops that are inherently determined by the iterative optimization process and the nature of the diffusion model in Alg. 1. Firstly, when updating the prediction data using the loss between the predicted and label data, the step size must not be too large, to avoid exceeding the model's effective data distribution, which could lead to artificial artifacts. Therefore, in order to achieve optimal optimization results, a progressive loop-based optimization approach is necessary. In addition, during the reverse diffusion process, each prediction at every time step needs to be re-noised before entering the next time step for further computation and optimization. This repetitive process of adding noise can disturb the already partially optimized data, causing a loss in the preservation of the relevant features of the label data. As a result, re-optimization at each time step is required. These factors inevitably affect the speed of the optimization process to some extent. Future research will focus on developing more efficient optimization strategies and improved denoising sampling methods to enhance performance.

## D  Large Language Model Usage Statement

In the preparation of this manuscript, a Large Language Model (LLM) was utilized strictly as a general-purpose editorial assist tool to enhance the presentation of the work.The application of the LLM was limited exclusively to the refinement of the written language. Specifically, its functions included: (1) Grammar and Mechanics Correction: Identifying and rectifying errors in syntax, punctuation, and spelling to ensure adherence to standard academic English. (2) Stylistic Polishing and Fluency Enhancement: Optimizing existing sentence structures and phrasing to improve overall textual clarity, coherence, and professional tone. Crucially, the LLM played no role in the intellectual contribution to this research. It was not employed for research ideation, the design of methodology, the execution of data analysis, or the formulation of core arguments, findings, or conclusions. As such, the LLM is not considered a contributor to the research effort. The authors collectively affirm and take full, exclusive responsibility for the entire content of this submission. This responsibility extends to all material, including any text generated or modified by the LLM that could be construed as plagiarism, academic misconduct, or the fabrication of facts.

## E  Ethics statement

We focus on fundamental computer vision tasks, e.g, style transfer, super-resolution, and deblurring. We are committed to upholding the highest standards of research integrity and social responsibility.

All datasets utilized in this study are publicly available benchmark datasets widely used within the academic community, and their usage complies with their respective terms of release. Crucially, our experiments rely solely on publicly accessible data. The research does not involve human subjects, nor does it entail the collection, use, or storage of any Personally Identifiable Information. Consequently, this work raises no concerns regarding individual privacy or data security.

The technological goals of this study—enhancing image quality and visual fidelity—are benign and application-neutral. Our methodologies and potential applications do not fall into areas that could lead to social harm, discrimination, or bias. We affirm that all authors have no financial or other forms of conflict of interest related to this research or the entities involved. All results and conclusions presented are derived from rigorous scientific methods, ensuring the objectivity and transparency of our findings.

## F  More Qualitative Results

We provide more qualitative results of our method in this section for deeper visual analysis and understanding. These qualitative results supplement the previously reported quantitative evaluation data, aiming to show the performance and effect of our model.

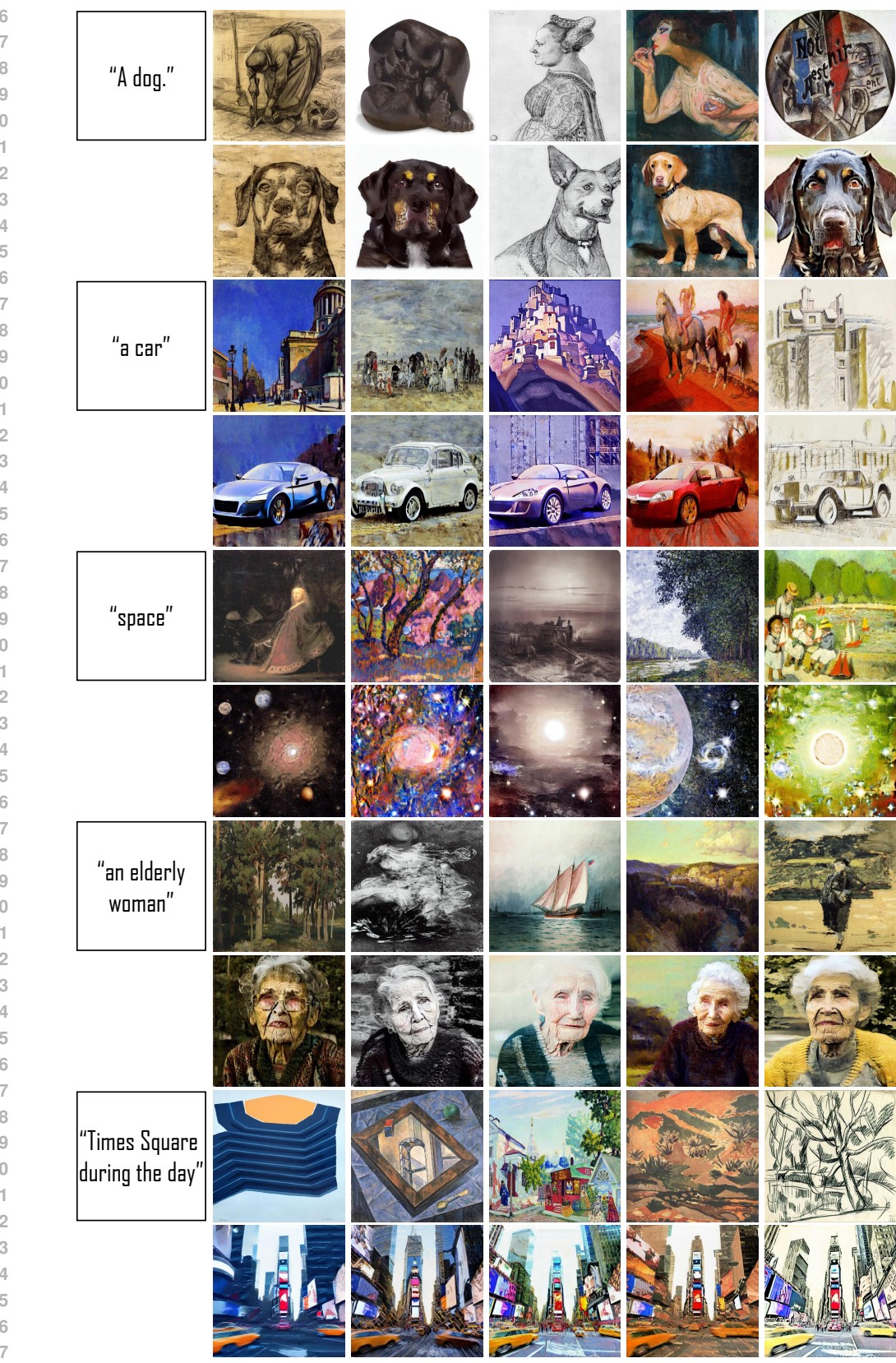

Figure 6: Qualitative results on the style transfer task. For each group, the top row shows the content and style references, and the bottom row is the resulting stylized output.

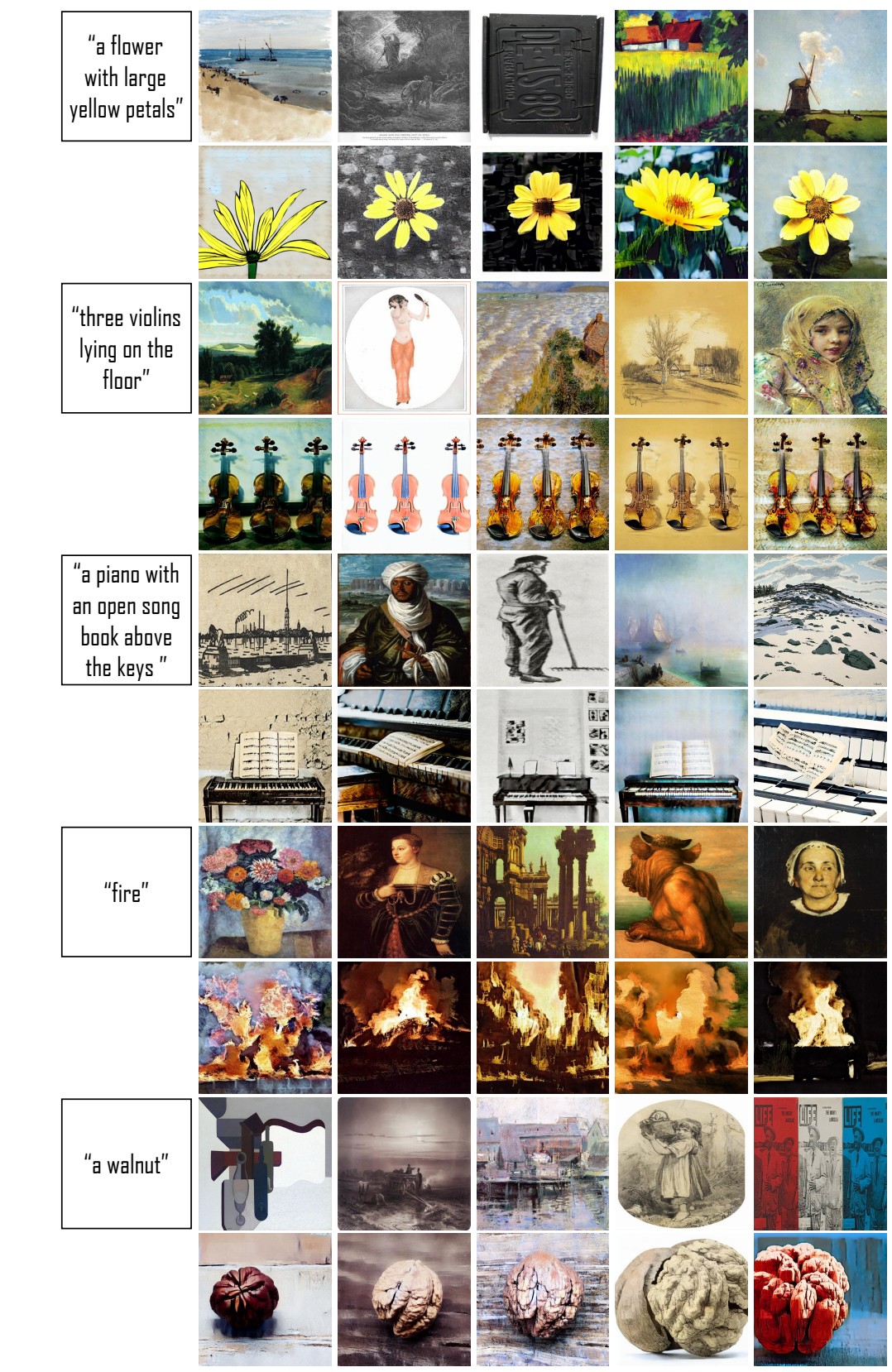

Figure 7: Qualitative results on the style transfer task. For each group, the top row shows the content and style references, and the bottom row is the resulting stylized output.

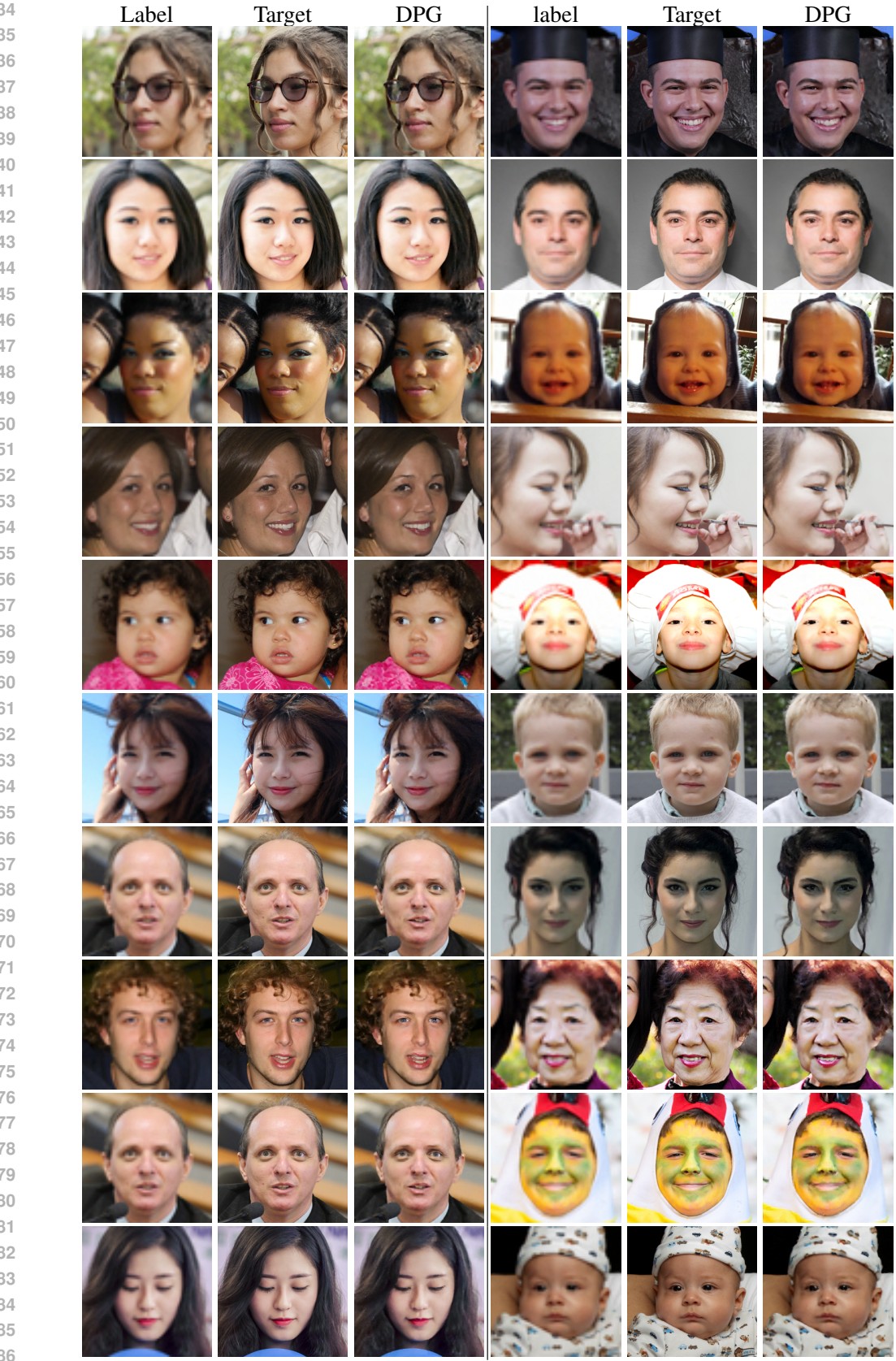

Figure 8: Qualitative results on the image super-resolution task.

| Label | Target | DPG | label | Target | DPG |
|-------|--------|-----|-------|--------|-----|

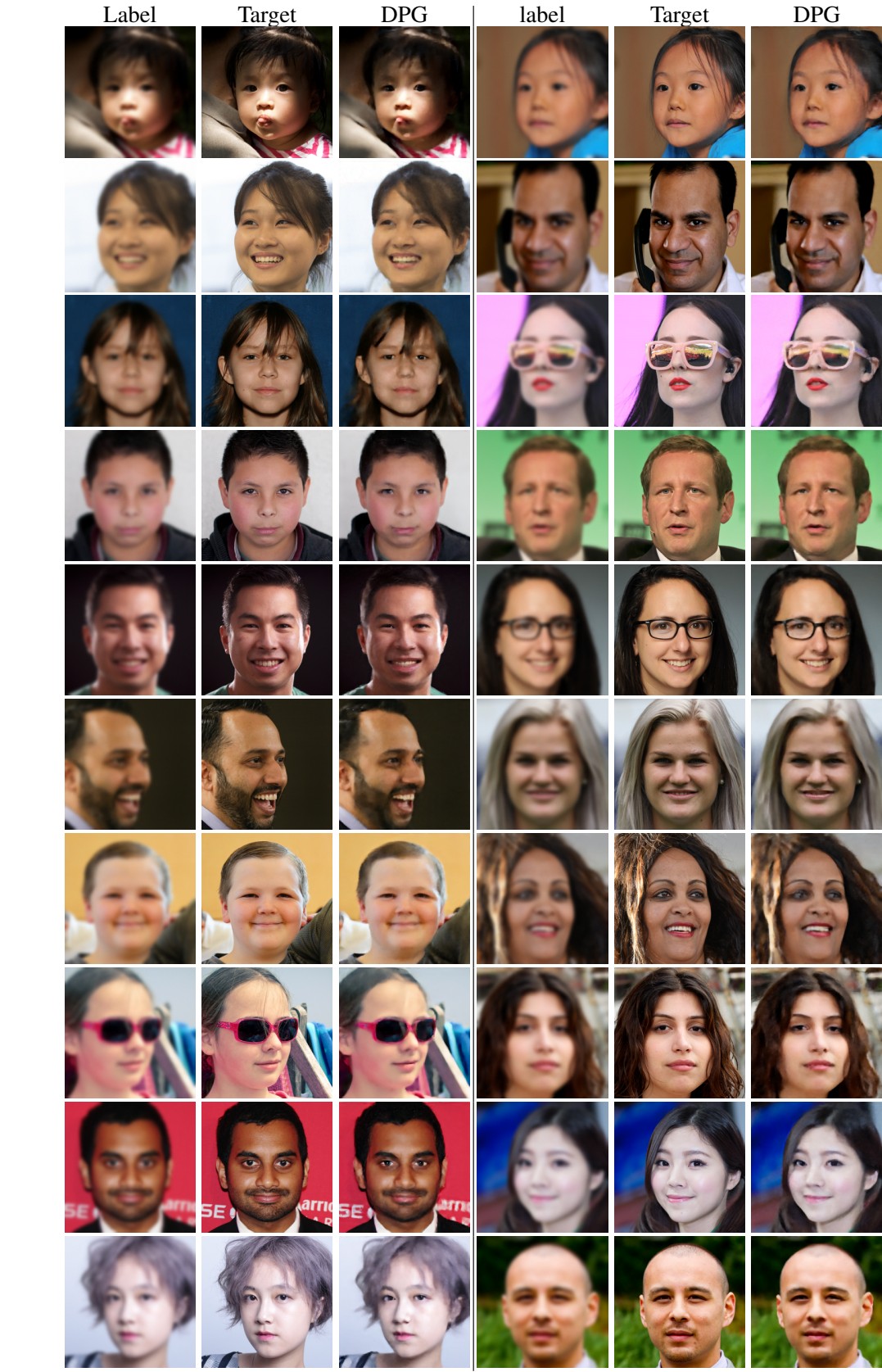

Figure 9: Qualitative results on the image deblurring task.

