# OpenReview forum: "DPG: Exploiting Data and Process Knowledge for Diffusion Guidance"
_ICLR.cc/2026/Conference — Submitted to ICLR 2026_

### Official Review · Reviewer_B5jh · 2025-10-31

**Soundness:** 2
**Presentation:** 2
**Contribution:** 3
**Rating:** 4
**Confidence:** 3

**Summary:**

This paper redefines style transfer as a weak-label guidance problem and image super-resolution / deblurring as degraded-label guidance. It argues that these existing methods are typically tailored to specific tasks, lack generalization and transferability, and often rely solely on loss-based optimization while neglecting underlying domain priors. To address this limitation, the authors propose a unified framework that leverages data knowledge inherent to each task and process knowledge extracted from the reverse diffusion process to solve image restoration and style transfer tasks jointly. They claim that unifying these two tasks is inherently challenging because style transfer depends on abstract style information, whereas restoration tasks use strong, explicit conditioning. Furthermore, they critique loss-gradient-based guidance methods, emphasizing that loss values may not accurately reflect image fidelity and are vulnerable to error propagation. Their method interpolates between the noised label prediction and the generated image from the reverse diffusion process at each step, while simultaneously enforcing label information through a loss computed between the predicted $x_0$ and the target $y$

**Strengths:**

- This paper addresses an important and timely problem: unifying weak-label (style transfer) and degraded-label (image restoration) tasks under a single diffusion-guided framework.
- The proposed approach, integrating data knowledge and process knowledge, is conceptually innovative and could provide new insights for handling tasks with imperfect or indirect supervision.

**Weaknesses:**

- The motivation for unifying two inherently distinct tasks, style transfer and image restoration, is not sufficiently substantiated. The authors (L99-L111) suggest potential knowledge transfer between the two, but no empirical evidence or references are provided. In fact, these tasks possess fundamentally different supervision properties: style transfer relies on weak, abstract cues, whereas restoration requires high-fidelity alignment with the reference. This discrepancy raises concerns about negative transfer when attempting to integrate both under a single framework. Including empirical evidence or prior works demonstrating positive transfer between these tasks would strengthen the argument.
- The inclusion of task-specific components, such as the label-processing function $M$ and hyperparameters like $\eta_1$, appears to contradict the paper’s claim of a unified framework. If these elements must be individually tuned per task, the true level of unification remains ambiguous.
- In L263, the authors briefly compare their approach with SDEdit, but the described mechanism, mixing the noised conditioning image during the diffusion process, closely resembles the method proposed in ILVR [a]. A proper citation and discussion of ILVR should therefore be included for completeness.

Minor Weaknesses

- The deblurring setup assumes a Gaussian blur kernel, which may be an overly restrictive assumption and limit general applicability.
- The paper seems to inconsistently use \citet and \citep throughout; all citations should be revised for consistency according to the chosen bibliography style.

[a] Choi, Jooyoung, et al. ILVR: Conditioning Method for Denoising Diffusion Probabilistic Models., CVPR 2021

**Questions:**

See weaknesses

---

### Official Review · Reviewer_Ybdq · 2025-10-31

**Soundness:** 2
**Presentation:** 2
**Contribution:** 2
**Rating:** 2
**Confidence:** 5

**Summary:**

The paper proposes DPG, which is a unified guideline framework for tasks with incomplete labels, covering both weak label settings (e.g., style transfer) and degraded label settings (e.g., super-resolution, denoising). The core idea is (1) to incorporate “data knowledge” by diffusing the (possibly processed) label itself and incorporating it into the early steps of the reverse diffusion process, and (2) to enforce “process knowledge” through a stepwise constraint, whereby each step of denoising improves the label alignment compared to the previous step. Experiments on style transfer and inverse problems show comparable qualitative and quantitative results.

**Strengths:**

- The proposed DFG can be used for stylization, SR, and de-noising, especially without changing the architecture. In addition, DFG can be easily applied to the existing reverse diffusion process.
- The paper distinguishes between “what to inject" (signal derived from the label), “when to inject” (early steps), and “how to keep progress" (stepwise constraint). This modular presentation makes the pipeline easy to understand.

**Weaknesses:**

- The authors redefine stylization as “weak label guidance” and classic inverse problems (e.g., super-resolution, denoising) as “degraded label guidance,” which differs from established usage in the computer vision literature. Using the available measurement/observation $y$ as a guide during the reverse diffusion process is already common practice in existing algorithms for both stylization and inverse problems. Therefore, this work cannot be considered the first to "unify" these approaches; such a formulation risks confusing the readership rather than clarifying the contribution.
- To capture the measurement/observation inforamtion, $z_{0|t}$ is refined  using eq. (9). This objective function $\ell_1 (z_{0|t},  y)$ is already frequently used to improve the performance of stylization and inverse problems. This cannot be considered a novelty of the proposed DPG.
- For process knowledge integration, $\alpha_\text{margin}$ would strongly influence the performance of tasks. Depending on other hyperparameters such as $T_1$, $\eta$, and $\gamma$, the quality of results may change. However, the authors do not provide any ablation studies. They just offer simple guidelines in the supplementary material. Based on the currently available information, the algorithm cannot be reproduced.
- The qualitative comparisons presented in the paper cannot confirm that the proposed DPG clearly outperforms existing algorithms. Furthermore, the image quality is not sufficient to compare the algorithms.

**Questions:**

- For the data knowledge integration, why are two linear interpolations fundamentally required in eq. (7) - (i) mixing the latent ($z_t$ and $\hat{c}_t$) and (ii) mixing the predicted noise (${\epsilon}(z_t, \cdot)$ and $\epsilon(c_t, \cdot)$? In other words, what stability principle makes both convex combinations necessary together, rather than one of them (or another mechanism) being sufficient?
- How high are the runtime and memory requirements of DFG compared to the standard reverse diffusion processes and existing algorithms?
- The ablation studies in relation to the hyperparameters

---

### Official Review · Reviewer_D3Cj · 2025-10-31

**Soundness:** 2
**Presentation:** 3
**Contribution:** 2
**Rating:** 4
**Confidence:** 3

**Summary:**

DPG is a training-free guidance method for “imperfect-label” problems: both weak labels (like a style image) and degraded labels (like low-res or blurred inputs). It first applies a simple task-specific pre-process to the label (e.g., upsampling or Wiener deblur). In the early denoising steps only, DPG injects “data knowledge” by blending that noised label with the current latent and by mixing its predicted score with the model’s normal score, giving the sampler scale-matched structural hints without hard constraints. At each step, a task loss nudges the current clean prediction toward the label (e.g., VGG/CLIP losses for style; re-degradation consistency for SR/deblur). Across the whole trajectory, DPG adds “process knowledge” via a margin term that enforces step-to-step improvement in the task loss, which suppresses zig-zagging and error accumulation.

**Strengths:**

- Novelty on label integration: forward-diffusing the label and blending it for additional integration of the label seems interesting. While I am not fully convinced why this should work well, it seems novel.
- Performance: compared to baselines, DPG seems to be superior on various tasks.

**Weaknesses:**

- Unfortunately, the paper format doesn't follow ICLR guidelines yet. The guidelines say as follows: ```Citations within the text should be based on the \texttt{natbib} package
and include the authors' last names and year (with the ``et~al.'' construct
for more than two authors). When the authors or the publication are
included in the sentence, the citation should not be in parenthesis using \verb|\citet{}| (as
in ``See \citet{Hinton06} for more information.''). Otherwise, the citation
should be in parenthesis using \verb|\citep{}| (as in ``Deep learning shows promise to make progress
towards AI~\citep{Bengio+chapter2007}.'').``` To follow this guideline, the first sentence of the introduction section should start with "In recent times, diffusion models (DMs) (Ho et......)". Unfollowing this ICLR format guideline makes it hard to read the paper.
- DPG introduces many hyperparameters, because it should determine 1) mixing weights, 2) early-step window, 3) margin, and 4) inner step sizes. If this parameter should be changed according to tasks, the author's main claim that DPG is a generalizable framework for weak-label and degraded-label tasks should not hold, because this means not a generalizable framework but an adjustable framework according to tasks.
- Justification depth: While intuitions are clear, the paper lacks a deeper analysis of line 250: "By adding noise and applying guidance, we let the model select the most relevant information for the task." My question is whether just mixing the noisy label is enough to change the model's behavior to select the most relevant information. In the rebuttal and discussion phase, I would like to discuss this with the author and revise the paper to correctly understand the reader. In current form, I am more towards that just mixing is not enough to enforce the model to choose relevant information only.

**Questions:**

- In Appendix, I found that the hyperparameters should be changed according to tasks. Is DPG sensitive to hyperparameter changes?

---

### Official Review · Reviewer_dEcG · 2025-11-01

**Soundness:** 3
**Presentation:** 2
**Contribution:** 2
**Rating:** 4
**Confidence:** 3

**Summary:**

This paper introduces DPG (Diffusion Process Guidance), a training-free framework for improving diffusion model guidance in imperfect-label tasks such as style transfer, super-resolution, and deblurring.
DPG combines data knowledge, by injecting diffused representations of imperfect labels into early denoising steps, and process knowledge, by enforcing progressive consistency across timesteps.
This unified approach refines diffusion trajectories without task-specific tuning, achieving better perceptual quality and fidelity across diverse restoration and translation tasks.

**Strengths:**

The paper presents a unified and training-free framework that creatively combines data- and process-level knowledge for diffusion guidance.
The idea of injecting diffused label information and enforcing progressive alignment offers a novel and generalizable formulation across multiple imperfect-label tasks.
Experiments are comprehensive and consistent, showing clear improvements in perceptual quality.
The paper is well-written and easy to follow, with solid motivation and clear methodological structure.

**Weaknesses:**

1. **Clarity and originality of the “unified framework” claim.**
   While the paper frames DPG as a unified, training-free guidance framework, several prior works have already explored similar directions. For example, *Manifold Preserving Guided Diffusion* (ICLR 2024) formulated a single training-free framework that covers super-resolution, deblurring, face-ID, CLIP, and style guidance—all without an explicit forward operator. Thus, the novelty of unification alone may be limited. The introduction dedicates considerable space to emphasizing potential advantages of such unification, but the paper would benefit from demonstrating concrete benefits (e.g., improved generalization across tasks or reduced hyperparameter tuning) rather than discussing them conceptually.

2. **Task-specific dependency of the mapping function ( M ).**
   Although the framework is described as general, it still requires defining a mapping function ( M ) to transform task-specific labels ( y ) into an image-like form. This step is not always straightforward and may vary considerably across tasks. In some inverse problems (e.g., phase retrieval or Fourier-domain tasks in *Diffusion Posterior Sampling*), such a mapping is not directly feasible, which somewhat limits the claimed generality of the approach.

3. **Missing discussion and comparison with closely related work.**
   The *Manifold Preserving Guided Diffusion* paper—arguably the most methodologically similar prior work—is not cited or discussed. That work also introduces an ( x_0 )-alignment loss with target information, conceptually close to Equation (9) here. Including a citation, a brief discussion, and possibly a quantitative comparison (e.g., in Fig. 4 or Table 1) would clarify the distinction between DPG and prior frameworks and make the contribution positioning clearer.

4. **Need for deeper analysis of the core components.**
   The main novelty appears to lie in the *Data Knowledge Integration (DKI)* and *Process Knowledge Integration (PKI)* modules, yet their effects are not analyzed in depth.
   – For **DKI**, injecting label information during early denoising is an intuitive and appealing idea, particularly since early predictions are often blurry and unstable. However, the ablation table (Table 2) seems inconsistent—entries (b) and (c) report significantly lower PSNR than the “w/o D” or “w/o P” variants, which might be a reporting error.
   – To help readers better appreciate DKI’s role, it would be valuable to visualize several denoising trajectories, comparing predicted ( x_0 ) with and without DKI. Such qualitative evidence would make the mechanism’s effect clearer and strengthen the empirical analysis.

**Questions:**

- Basically, all points mentioned in the Weaknesses section.
- The role of ( $\alpha_{\text{data}}$ ) is intuitively clear—it controls how much information is drawn from the label—but the meaning of ( $\gamma_{\text{data}}$ ) is less so. It seems analogous to a CFG-style guidance term interpolating between predictions from label-mixed and original latents. Why is this additional guidance needed if prediction from the label-mixed latent alone is already available? What new information is obtained from extrapolating between the two predictions? Given that this doubles the computational cost, please justify the necessity of this step and, if possible, provide qualitative or quantitative results across several key tasks for different ( $\gamma_{\text{data}}$ ) values.

If I have misunderstood this mechanism, I would welcome clarification. My rating could increase accordingly.

---

### Meta-Review · Area_Chair_Lmy9 · 2025-12-25

**Summary:**

All reviewers voted against accepting the paper for concerns on originality, missing technical analysis, comparison to baselines, lack of reproducibility, and actual lack of unifying guidance. The authors provided no argument against these concerns. I encourage the authors to resubmit the paper in the future with these concerns in mind.

**Reviewer Concerns:**

There were no answers provided by the authors.

**Reviewer Scores:**

No change, since no rebuttal.

---

### Decision · Program_Chairs · 2026-01-26

Reject